# Hepatocellular Toxicity of Paris Saponins I, II, VI and VII on Two Kinds of Hepatocytes-HL-7702 and HepaRG Cells, and the Underlying Mechanisms

**DOI:** 10.3390/cells8070690

**Published:** 2019-07-09

**Authors:** Wenping Wang, Yi Liu, Mingyi Sun, Na Sai, Longtai You, Xiaoxv Dong, Xingbin Yin, Jian Ni

**Affiliations:** 1School of Chinese Materia Medica, Beijing University of Chinese Medicine, Beijing 100102, China; 2Beijing Research Institute of Chinese Medicine, Beijing University of Chinese Medicine, Beijing 100102, China

**Keywords:** Paris saponins I, II, VI, and VII, cell apoptosis, Bcl-2 proteins, ROS generation, activation of death receptor

## Abstract

*Rhizoma paridis* is a popularly-used Chinese medicine in clinics, based on the pharmacodynamic properties of its saponin components. The four main saponins in *Rhizoma paridis* are designated saponins I, II, VI, and VII. At present, much attention is focused on the anticancer effect of *Rhizoma paridis* which is manifested in its cytotoxicity to various cancer cells. The purpose of this study was to investigate the hepatocellular toxicities of the four saponins in *Rhizoma paridis* and the relative intensities of their cytotoxic effects. It was found that the four saponins were cytotoxic to two types of hepatocytes-HL-7702 and HepaRG cells. The cytotoxicities of the four saponins to the two cell models were compared. One of the most cytotoxic saponins was *Rhizoma paridis* saponin I (PSI). This was used to determine the mechanism of hepatocellular toxicity. Results from MTT assays demonstrated that the four saponins induced apoptosis of the two hepatocyte models in a dose-dependent and time-dependent manner. In addition, fluorescent 4′,6-diamidino-2-phenylindole (DAPI) staining was used to observe the morphological changes of HepaRG cells after saponin administration. Further, as the concentration increased, PSI-induced lactate dehydrogenase (LDH) release from HepaRG cells increased gradually. In addition, PSI enhanced the levels of reactive oxygen species (ROS) and blocked the S and G2 phases of the cell cycle in HepaRG cells. A western blot indicated that PSI upregulated the protein expression levels of p53, p21, and Fas. Furthermore, the PSI-induced changes in the p53 protein increased the Bax/bcl-2 ratio, resulting in enhancement of the release of mitochondrial cytochrome c, activation of caspases-3, -8, and -9, poly-ADP ribose polymerase (PARP), and ultimately apoptosis. Increased Fas protein activated caspase-8, which led to the activation of caspase-3 and its downstream PARP protein, resulting in cell apoptosis. These results indicate that PSI induced apoptosis in HepaRG cells through activation of ROS and death receptor pathways. The results obtained in this study suggest that the hepatocellular toxicity of saponins in *Rhizoma paridis* should be considered during the clinical application of this drug. In addition, they provide a reference for future anti-cancer studies on *Rhizoma paridis*.

## 1. Introduction

*Rhizoma paridis,* a liliaceous plant of the Yunnan Paris or the dried rhizome of the seven-leaf flower, is a frequently used traditional Chinese medicine in clinics due to its numerous pharmacological effects [1,2]. Recently, some studies reported that an excess dose of *Rhizoma paridis* produced liver injury due to its saponin components [3,4,5,6,7]. However, the hepatotoxicity of various saponins in *Rhizoma paridis* and their mechanism of action have not been studied. Most of the research on the saponins in *Rhizoma paridis* has focused on their inhibitory effects on the growth of various tumor cells [8,9,10,11,12]. However, the present study investigated the hepatocellular toxicity of the four major saponins in *Rhizoma paridis.* The data obtained revealed that the four major saponins (Paris saponins I, II, VI, and VII) in *Rhizoma paridis* (Figure 1) exerted different levels of cytotoxicities on two kinds of liver cells. 

HepaRG cells retain the characteristics of many primary hepatocytes, including the expression of critical metabolism enzymes, drug transporters, and nuclear receptors. Recently, the hepatotoxicity [13,14,15] associated with traditional Chinese medicine has attracted significant attention. Some researchers [16,17] found that HepaRG cells can be used to assess the hepatocellular toxicity induced by Chinese medicine after repeated treatments or acute treatment. On the other hand, HL-7702 cells isolated from normal human livers are normal hepatocytes. Compared to hepatocellular carcinoma, HL-7702 cells possess different ultrastructures and are the most frequently used model for cytotoxicity studies in vitro. 

Drug-induced cell apoptosis is an important index of its toxicity. Apoptosis is defined as programmed cell death that is necessary for normal physiological functions [18,19,20]. Bcl-2 is a typical anti-apoptotic protein, while Bax is a pro apoptotic protein [21,22]. An up-regulated Bax/Bcl-2 ratio leads to the activation of caspases, which are the ultimate executors of apoptosis [23,24]. Caspase-9, a promoter of apoptosis [25], undergoes cleavage activation under the influences of a mitochondrial programmed death signal and the caspase-8 mediated death receptor pathways [26,27]. Both caspases cleave and activate the executioner enzyme caspase-3, eventually resulting in apoptotic cell death [28,29]. In previous studies, two signal pathways of cell apoptosis were identified: the extracellular pathway (death receptor pathway) and the intracellular pathway (the mitochondria pathway) [30,31]. 

The aim of this study was to investigate the hepatocellular toxicities of four saponins with various pharmacodynamic properties in *Rhizoma paridis* and to determine their IC_50_ values and compare their hepatocellular toxicities. This study aimed to provide reference data for future anti-tumor studies on these saponins to find out if they kill cancer cells without producing hepatocellular toxicity. Moreover, this study was carried out to lay a foundation for the development of safer and more effective anti-cancer drugs from Paris saponins. More specifically, the data revealed that the four saponins were cytotoxic to HepaRG cells and HL-7702 cells. In the process of cell culture, it was found that HepaRG cells (a new type of cell for evaluating hepatocellular toxicity) were easier to grow and reproduce than HL-7702 cells. Moreover, Paris saponin I (PSI), a representative and the most important saponin in *Rhizoma paridis,* has extensive antitumor effects [32,33]. Therefore, in the next study, HepaRG cells were taken as the cell model and PSI was selected as the model drug to further investigate the mechanism of the hepatocellular toxicity of saponins in *Rhizoma paridis.* It was found that reactive oxygen stress pathways play an important role, and it was confirmed that the PSI-induced apoptosis of HepaRG cells occurred through mechanisms involving the regulation of proteins associated with reactive oxygen stress pathway and the Fas death pathway. These are novel findings. The results provide a warning against the inappropriate clinical use of saponins from *Paris polyphylla*. Moreover, the hepatocellular toxicities of saponins from *Rhizoma paridis* should be investigated further with respect to the suitability of these saponins as anticancer drugs.

## 2. Materials and Methods

### 2.1. Drugs and Reagents

Paris saponin I (batch no.111590-201604, purity >97.6%), Paris saponin II (batch no.111591-201604, purity >97.4%), Paris saponin VI (batch no.111592-201604, purity >97%), and Paris saponin VII (batch no. 111593-201604, purity >97.6%) were purchased from the Chinese Food and Drug Inspection Institute (Beijing, China). Penicillin, streptomycin solution, and 0.25% trypsin were acquired from Corning (Corning, NY, USA). Dulbecco’s modified Eagle’s medium (DMEM), Roswell Park Memorial Institute (RPMI) 1640 medium, fetal bovine serum (FBS), and dimethyl sulfoxide (DMSO) were acquired from Solarbio (Beijing, China). 3-[4,5-Dimethylthiazol-2-yl]-2,5 diphenyl tetrazolium bromide (MTT), DAPI (4’,6-diamidino-2-phenylindole), lactate dehydrogenase (LDH), mitochondrial membrane potential (MMP) assay kit with JC-1; 2,7-dichlorofluorescein diacetate (DCFH-DA), Annexin V-fluorescein isothiocyanate (FITC) apoptosis measurement kit, cell cycle, and apoptosis analysis kits were purchased from Beyotime (Nanjing, China). The antibodies of cleaved caspase-8 (#ab25901) were obtained from Abcam (Shanghai, China) (#ab25901). Other antibodies including Bax (#5023T), Bcl-2 (#15071), cytochrome c (#4280T), p53 (#2524T), p21 (#2947T), cyclin A (#4656T), CDK2 (#2546T), Fas (#4233), PARP (#9542T), cleaved caspase-3 (#9661T), and cleaved caspase-9 (#9501T), were bought from Cell Signaling Technology (Beverly, MA, USA). 

### 2.2. Cell Culture Conditions

The HepaRG cell line was acquired from Guangzhou Jenniobio Biotechnology Co., Ltd., China. This cell line was added to an RPMI-1640 culture medium containing 10% FBS and 1% penicillin and streptomycin solution and then cultured in a cell incubator at 37 °C and 5% CO_2_. Paris saponins I, II, VI and VII were dissolved in dimethyl sulfoxide (DMSO) and the initial concentration was 40 uM. The concentration of the working solution of DMSO did not exceed 0.1% in cell culture experiments [32]. The HL-7702 cells were obtained from China Infrastructure of Cell Line Resources and added to Dulbecco’s modified Eagle’s culture medium containing 10% FBS and 1% penicillin–streptomycin solution. 

### 2.3. Comparison of Cytotoxicities of the Four Saponins in Rhizoma Paridis on HL-7702 Cells

The HL-7702 cells were seeded in 96-well plates at a density of 5 × 10^3^ cells/well. After 24 h, the cells were incubated with different concentrations of Paris saponins I, II, VI, and VII (0, 0.25, 0.5, 1, 2, 4, and 8 uM). After 24, 48, and 72 h incubation, cell viability was determined using MTT assay. The control group of cells had 0.1% DMSO in place of saponins. The treated cells were then cultured in MTT working solution (0.5mg/mL) at 37 °C for 4 h. Subsequently, the medium containing MTT was removed [34], and the resultant formazan crystals were solubilized in 150 uL DMSO. The absorbance of the resultant solution was read at 570 nm in a microplate reader (Thermo, Multiskan GO, USA). The relative viability of the treated cells was expressed as percentage of the control untreated cells. The IC_50_ values were calculated based on the percentage of cell survival at different time points with the SPSS version 17 software (SPSS Company, Chicago, IL, USA). The smaller the IC_50_ value, the greater the hepatic cytotoxicity. 

### 2.4. Comparison of Cytotoxicities of the Four Saponins in Rhizoma Paridis to HepaRG Cells

HepaRG cells were incubated with different concentrations of Paris saponins I, II, VI and VII (0, 0.25, 0.5, 1, 2, 4 and 8 uM) for 24 h. Thereafter, cell viability was determined with the same method used for HL-7702 cells. Then, IC_50_ values were calculated for the different saponins 24 h after administration to HepaRG cells. The IC_50_ values of the saponins 24 h after their administration were compared to determine their relative toxicities to the two types of hepatocytes.

### 2.5. LDH Assay

The HepaRG cells (6 × 10^5^ cells/mL) were treated with serial concentrations of PSI (0, 1.25, 2.5, 5, and 7.5 uM) for 24 h. Then, absorbance was read at 490 nm in a microplate reader (Thermo, Multiskan, GO, USA). The amount of LDH released under different dosages was calculated.

### 2.6. DAPI Staining

Nuclear morphological changes in the cells after saponin administration were observed using DAPI staining with fluorescent dye. The HepaRG cells were seeded at a density 6 × 10^5^ cells/mL and treated with serial concentrations of PSI for 24 h. Thereafter, the cells were washed once with PBS, and then fixed in PBS for 15 min at room temperature with 4% paraformaldehyde. The fixed cells were washed with PBS and stained with 2.5 μg/mL DAPI solution at room temperature for 10 min [35]. Thereafter, the cells were washed two more times with PBS, and the apoptotic cells were identified through examination under an inverted Olympus IX71 fluorescence microscope (Tokyo, Japan).

### 2.7. Determination of the Effect of PSI on Apoptosis and Necrosis

Flow cytometry was used to evaluate membrane asymmetry externalization and membrane integrity. The HepaRG cells were treated with different concentrations of PSI for 24 h. Untreated cells served as control. Apoptosis was analyzed with an AnnexinV/PI double-staining assay using the Annexin V-FITC assay kit. Following PSI treatment, the cells (6 × 10^5^ cells/mL) were washed with PBS and then re-suspended with 295 μL binding buffer. Then, Annexin V-FITC (5 μL) and propidium iodide (10 μL) were added. After a 15 min incubation period at room temperature in the dark, the cells were subjected to flow cytometric analysis [36].

### 2.8. Determination of Effect of PSI on Intracellular ROS Levels

The effect of PSI on generation of ROS was determined using an oxidant-sensitive fluorescent probe DCFH-DA [37]. In order to test the effect of PSI on concentration on ROS levels in HepaRG cells, the cells were evenly seeded in the 96-well plates (6 × 10^5^ cells/mL) and were treated with PSI at concentrations of 0, 1.25, 2.5, 5, and 7.5 uM for 24 h. Subsequently, the cells were incubated in the dark with 10 uM DCFH-DA at 37 °C for 10 min. Esterified DCFH-DA is capable of penetrating the cell membrane, and in the cell, it is de-acetylated by intracellular esterase. The resulting compound, i.e., DCFH, is trapped into the cells where it reacts with ROS to produce an oxidized fluorescent compound 2,7-dichlorofluorescein (DCF). The fluorescence intensity of DCF, which was determined flow cytometrically, reflects the amount of ROS in the cells.

### 2.9. Determination of the Effect of PSI on Mitochondrial Transmembrane Potential (∆ψ)

Mitochondrial transmembrane potential (∆ψ) was evaluated using JC-1, a mitochondria-specific lipophilic cationic fluorescence dye, to which mitochondrion is selectively permeable. The dye exists as a green fluorescent monomer at low mitochondrial membrane potential, but it is transformed to a red fluorescent polymer at high membrane potential. The HepaRG cells were plated in 6-well culture plates at a density of 6 × 10^5^ cells/well and cultured for 24 h. Different concentrations of PSI were added and incubated for 24 h, after which JC-1 working solution (5 μg/mL) was added. The cells were incubated for 20 min at 37 °C in the dark. Following re-suspension in PBS, the HepaRG cells were analyzed flow cytometrically.

### 2.10. Cell Cycle Analysis

The cells were planted in 6-well plates at a density of 6 × 10^5^ cells/well overnight to make the cells achieve adherence. Then, they were treated with PSI at concentrations of 0, 1.25, 2.5, 5, and 7.5 uM for 24 h. Thereafter, the cells were fixed with ice-cold 70% ethanol and kept overnight at a temperature of 4 °C. Following fixation, the cells were washed with PBS and suspended in 1mL PI staining reagent (50 mg/mL) supplemented with 100 μg/mL RNase (a staining buffer solution) for 30 min at 37 °C in the dark. The fluorescence was measured in fluorescence channels FL3 (488 nm excitation and 585/42 nm emission for PI).

### 2.11. Assay of Effect of PSI on Expressions of Relevant Proteins Using Western-Blot

The cells were seeded in 6-well plates at a density of 6 × 10^5^ cells/well for 24 h, after which they were treated with 0, 2.5, and 5 uM PSI for 24 h. After incubation, the cell pellet was suspended in an ice-cold cell extraction buffer for 30 min, and then lysed in a sample buffer (50 mM Tris, pH 7.4 [31,32], containing 50 mM NaCl, 1% Triton X-100, 1% sodium deoxycholate, and 0.1% SDS). Cell lysates were centrifuged at 4 °C for 10 min with 15,777 rpm. The total protein content of the lysate was assayed using the bicinchoninic acid (BCA) protein assay kit (Beijing Beyotime Institute of Biotechnology, Beijing, China). Subsequently, the mitochondrial and cytosolic fractions were separated with the ProteoExtract Cytosol/Mitochondria Fractionation Kit (Millipore, Billerica, MA, USA). Equal amounts of protein (50–100 µg) were separated on a 10% sodium dodecyl sulfate-polyacrylamide gel electrophoresis (SDS)-polyacrylamide gels. Next, the separated proteins were transferred to a polyvinylidene fluoride (PVDF) membrane (Pall, New York, NY, USA). Non-specific binding was blocked by incubation with 5% skim milk in Tris-buffered saline containing 25 mM Tris, 150 mM NaCl, 0.1% Tween 20 (pH 7.4) for 1 h. The membrane was then incubated with appropriate primary antibodies at the indicated dilutions: Fas (1:1000), Bcl-2 (1:1000), Bax (1:1000), cleaved caspase-3 (1:1000), cleaved caspase-8 (1:1000), cleaved caspase-9 (1:1000), cyclin A (1:1000), cyclin E (1:1000), CDK2 (1:1000), cytochrome c (1:500), p21 (1:1000), p53 (1:1000), cleaved PARP (1:1000), and β-actin (1:5000). This was followed by incubation with secondary antibodies (1:5000 dilution) at room temperature for 1 h. The blot signals were detected with an electrochemiluminescence (ECL) western blotting detection reagent (Pierce, Appleton, WI, USA). 

### 2.12. Statistical Analysis

Results are expressed as mean ± standard deviation (SD) of three independent assays. Statistical analysis was done using a One-Way ANOVA analysis and Least Significant Difference (LSD) test using the SPSS software (17.0). Statistical significance was fixed at *p* ˂ 0.05. 

## 3. Results

### 3.1. Comparison of Cytotoxicities of the Four Saponins of Rhizoma Paridis on HL-7702 Cells

The viability of HL-7702 cells was determined 24, 48, and 72 h after each of the four saponins was administered. The results showed that the four saponins had significant effects on the viability of the HL-7702 cells. The cell viability of the treated cells was significantly lower than that of the control cell in a time- and dosage-dependent manner. As shown in Figure 2A, the viability of HL-7702 cells gradually decreased with an increasing concentration of PSI (from 0.25 to 8 uM) and also the duration of PSI exposure (24, 48, and 72 h). Through statistical calculation, we obtained the IC50 values of different saponins at different times after administration (Figure 2B). The order of toxicities of the four saponins to HL-7702 cells was: Paris saponin I ≈ Paris saponin VII > Paris saponin II > Paris saponin VI. 

### 3.2. Cytotoxic Effect of the Four Saponins on HepaRG Cells

The results showed that the four main saponins in Rhizoma Paris exerted toxic effects on HepaRG cells. As shown in Figure 3A, the viability of HepaRG cells gradually decreased with an increase in PSI concentration from 0.25 to 8 uM. The IC_50_ values after 24 h of administration are shown in Figure 3B. When the IC_50_ values were compared, the order of toxicity of the four saponins to HepaRG cells was: Paris saponin VII > Paris saponin II > Paris saponin I > Paris saponin VI. Upon comparing the data obtained 24 h after the two cells were treated with the four saponins, a consistent pattern was observed. Paris saponin VII produced the highest cytoxicity in the HepaRG cells, while the lowest cytotoxicity was produced by Paris saponin VI. The cytotoxicities of Paris saponin I and Paris saponin II were between those of Paris saponin VI and Paris saponin VII.

### 3.3. LDH release and DAPI Staining

Lactate dehydrogenase (LDH) is located in the cytoplasm. Its presence in the extracellular medium can be used as an index of cell membrane integrity. With an increase in dose, DAPI staining clearly revealed aggregation of chromatin and nuclear fragmentation in HepaRG cells. (Figure 4A). In addition, PSI treatment for 24 h caused LDH leakage in a dose-dependent manner (Figure 4B).

### 3.4. Effect of PSI on the Apoptosis and Necrosis of HepaRG Cells

Next, an investigation was carried out to see whether PSI’s cytotoxicity to HepaRG cells was associated with the induction of apoptosis and necrosis. Annexin-V/PI double staining was used after 24 h of PSI administration (Figure 5A). The results in Figure 5B show that the early apoptotic cells were 8.13% at 1.25 uM, 14.93% at 2.5 uM, 34.33% at 5 uM, and 48.67% at 7.5 uM PSI. However, there was only 7.97% early apoptotic cells in the absence of PSI. Moreover, the late apoptotic and necrotic cells were 3.53, 6.13, 25.50, and 34.83% at PSI concentrations of 1.25, 2.5, 5, and 7.7 uM PSI, respectively, with only 2.37% late apoptotic cells in control untreated cells. The viable cells decreased from 88.47 to 15.60% with an increase in PSI concentration. In summary, as the PSI concentration increased from 1.25 to 7.5 uM, the viable cells gradually decreased while the early and the late apoptotic cells gradually increased.

### 3.5. Effects of PSI on ROS Generation

It has been shown that the generation of ROS damages DNA and proteins and leads to apoptosis. Thus, the generation of ROS in HepaRG cells was determined with a DCFH-DA assay. Results (Figure 6A,B) showed that after 24 h treatment with PSI, there was a dose-dependent increase in ROS content, up to 204.96% when the PSI concentration was 5 uM. However, there was a downward trend in ROS content at higher drug concentrations. To sum up, the results suggest that PSI plays a crucial role in inducing apoptosis by inducing ROS production and oxidative stress in HepaRG cells. 

### 3.6. Effect of PSI on MMP and Cytochrome c Released from HepaRG Cells

The effect of PSI on the mitochondrial membrane potential of HepaRG cells was determined using JC-1 staining. As shown in Figure 7A, the mitochondrial membrane potential decreased after incubation with graded concentrations of PSI for 24 h (Figure 7B). Thus, PSI increased mitochondrial membrane permeability and caused mitochondrial dysfunction, suggesting that PSI-induced apoptosis of HepaRG cells involves the mitochondrial pathway. It is known that a loss of MMP initiates the release of cytochrome c into the cytosol, which activates a caspase cascade, ultimately resulting in apoptosis [9]. To determine whether PSI elicited the release of cytochrome c to the cytosol, the levels of cytochrome c in the mitochondria and cytoplasm of the control group and administration groups were measured. Western blot results (Figure 7C,D) showed that the cytochrome c content of the cytoplasm significantly increased but was reduced in the mitochondria after treatment with PSI for 24 h.

### 3.7. Effect of PSI on Cell Cycle Distribution

Following treatment with PSI (0, 1.25, 2.5, 5, and 7.5 uM), incubation for 24 h, and fixation overnight in 70% cold ethanol, the cell cycle was analyzed in HepaRG cells. As shown in Figure 8A,B, PSI treatment resulted in an accumulation of cells in the S phase and G2/M phase, accompanied by a reduction of cells in G0/G1 phase, relative to the untreated group. The percentage of HepaRG cells in the S phase and G2/M phase rose from 25.43% ± 3.91% to 36.48% ± 0.98%, and from 3.58% ± 1.82% to 8.83% ± 1.36%, respectively. Then, the expression levels of the relevant proteins involved in the S and G2/M phase progression were measured using a Western blotting assay. The results showed that the expressions of p53, p21, and cyclin E proteins were significantly increased, while the expressions of cyclin A and CDK2 decreased in HepaRG cells after the treatment of PSI (Figure 8C,D). These results revealed that PSI inhibited cell cycle progression by changing the expression of cell cycle regulators. 

### 3.8. Effect of PSI on the Expression of Apoptosis-Related Proteins in HepaRG Cells

In order to elucidate the underlying mechanisms of the hepatocellular toxicity of PSI, the expressions of apoptosis-related proteins in PSI-treated HepaRG cells were investigated. As shown in Figure 9A,B, PSI induced dose-dependent increases in expressions of Bax, Fas, caspase-3, caspase-8, caspase-9, and PARP, while decreasing the expression levels of Bcl-2 (*p* < 0.05). Thus, Fas, a typical death receptor, was upregulated by PSI administration. In addition, PSI significantly promoted the expression of pro-apoptotic Bax and blocked the expression of anti-apoptotic Bcl-2 in a dose-dependent manner, thereby leading to an increased Bax:Bcl-2 ratio and eventually to apoptosis. Furthermore, PSI treatment apparently influenced the expression levels of death signals, thereby increasing the cleavage of caspases-3, 8, and 9. These results suggest that PSI induced hepatocyte apoptosis through the endogenous and exogenous pathways of apoptosis, both of which are dependent on the activation of caspases.

## 4. Discussion

*Rhizoma paridis* is a traditional and important medicinal Chinese herb, which has attracted significant research interest in recent years. Its main bioactive components are four kinds of saponins: Paris saponins I, II, VI, and VII [38,39]. The saponins in Paris saponins have many pharmacological properties, such as anti-tumor [40,41], hemostatic [42], anti-bacterial [43], anti-inflammatory [44], angiogenesis-inhibitory [45] and immune-regulatory [46] effects. However, in recent years, some studies have reported the in vivo hepatotoxicity of Paris saponins [3,4,5,6,7]. The purpose of the present study was to investigate the hepatotoxicity of the four main saponins from *Rhizoma paridis* in vitro and to unravel the cellular and molecular mechanisms involved. This aspect of research on saponins from *Rhizoma paridis* has not been reported in the literature.

In this study, two kinds of hepatocytes-HepaRG cells and HL-7702 cells were used as model cells, while Paris saponins I, II, VI, and VII from *Rhizoma paridis* were used as model drugs to study the hepatocellular toxicity of *Rhizoma paridis* saponins. The four saponins promoted apoptosis of HepaRG cells and HL-7702 cells. The hepatic cytotoxicity of Paris saponin VII was the strongest, while the hepatic cytotoxicity of Paris saponin VI was relatively weak. Since Paris saponin I is one of the most important saponins in Paris polyphylla, it was chosen as a representative drug to study the mechanism of hepatocellular toxicity.

Paris saponin I, originally derived from *Rhizoma paridis*, has a wide range of pharmacological effects, the most significant of which is its cytotoxic effect against a number of cancer cells [47,48]. Series of experiments were carried out to investigate the cytotoxic effect and mechanism of action of PSI on HepaRG cells. Through preliminary experiments, the range of PSI concentrations was determined to be 1.25, 2.5, 5, 7.5 uM. Assays of LDH revealed that PSI compromised the integrity of the HepaRG cell membrane, leading to LDH release into the cytoplasm. Morphological changes in the nucleus were observed through DAPI staining with fluorescent dye. Moreover, PSI strongly induced HepaRG cell apoptosis in a dose-dependent manner, as shown in Annexin V/PI staining experiments, suggesting that the toxic effects of PSI on HepaRG cells are mediated through the induction of apoptosis.

The ROS-oxidative stress pathway is a very important pathway that leads to apoptosis, and it has been widely studied [49,50,51]. The results (Figure 10) obtained in this study revealed that PSI affected the mitochondrial membrane’s integrity, enhanced ROS levels, released cytochrome c, and reduced ∆ψ [52], indicating that PSI induced oxidative stress and damage to cell membrane integrity in HepaRG cells. In addition, PSI arrested the cell cycle at the S and G2 phases. Subsequently, the expression levels of proteins involved in S and G2/M phase progression were measured. The expression levels of p53 and p21 were upregulated, while other cyclin proteins, i.e., cyclin A, cyclin E, and CDK 2, were decreased after treatment with PSI for 24 h. In all, PSI induced apoptosis through P53 expression [53], alteration of Bcl-2/Bax ratio, promotion of the release of cytochrome c from mitochondria, and activation of procaspase-9 and procaspase-3, all of which led to apoptosis [54]. 

In addition to the ROS-oxidative stress pathway, PSI also induced HepaRG cell apoptosis through the death receptor pathway. This was evident in the upregulation of Fas protein expression, and the activation of initiator caspases (caspase-8 and caspase-3), resulting in apoptotic cell death. In summary, PSI-induced apoptosis was executed through activation of ROS stress and death receptor pathways, both of which converge in the cascade activation of caspase proteases. The significance of these findings involves two aspects. Firstly, this study has established that the four saponins in *Rhizoma paridis* exert strong hepatocellular toxicities. Since *Rhizoma paridis* is a popular Chinese medicine in clinics, its potential hepatocellular toxicity deserves very serious attention. Secondly, Paris saponins have been widely reported as anticancer drugs. However, this study found that Paris saponins exert high toxicity to hepatocytes as well as to cancer cells. Therefore, the use of Paris saponins as anti-cancer drugs is questionable. In anti-tumor studies of the four saponins used in this study, the doses should perhaps be controlled within a range that is non-toxic to hepatocytes. These doses are clearly indicated in this study. 

## Figures and Tables

**Figure 1 cells-08-00690-f001:**
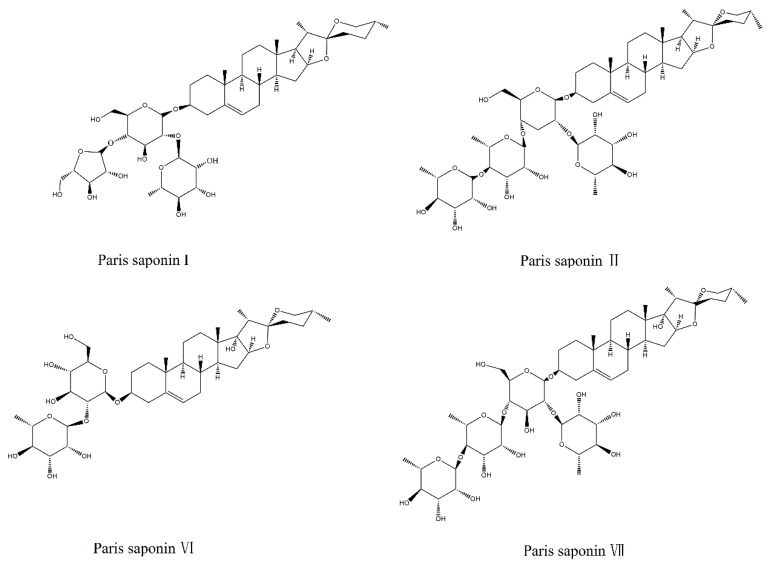
Chemical structures of Paris saponins I, II, VI, and VII.

**Figure 2 cells-08-00690-f002:**
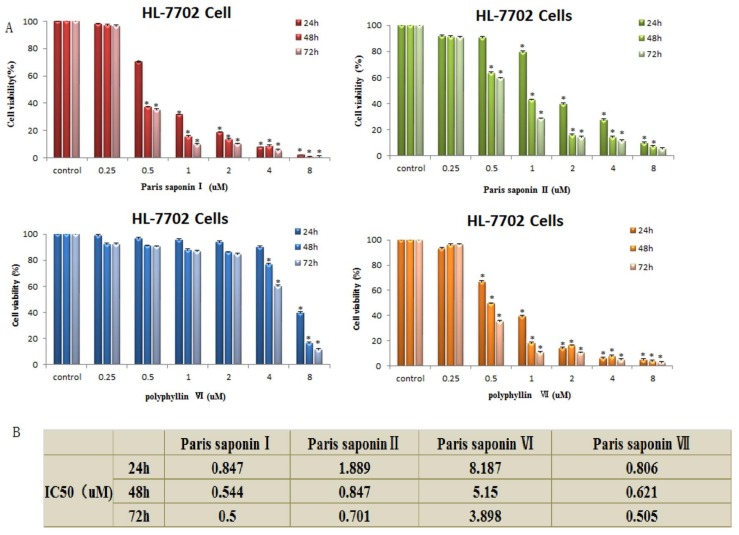
Cytotoxic effects of the four saponins on HL-7702 cells. (**A**) Cell viability of HL-7702 cells determined using an MTT assay. The data are presented as mean ± SD. * *p* < 0.05, compared with the vehicle-treated group. (**B**) IC_50_ values obtained by analysis of the cell viability data of the HL-7702 cells after treatment with the 4 saponins for 24, 48, and 72 h. The analysis was done using the SPSS17 software.

**Figure 3 cells-08-00690-f003:**
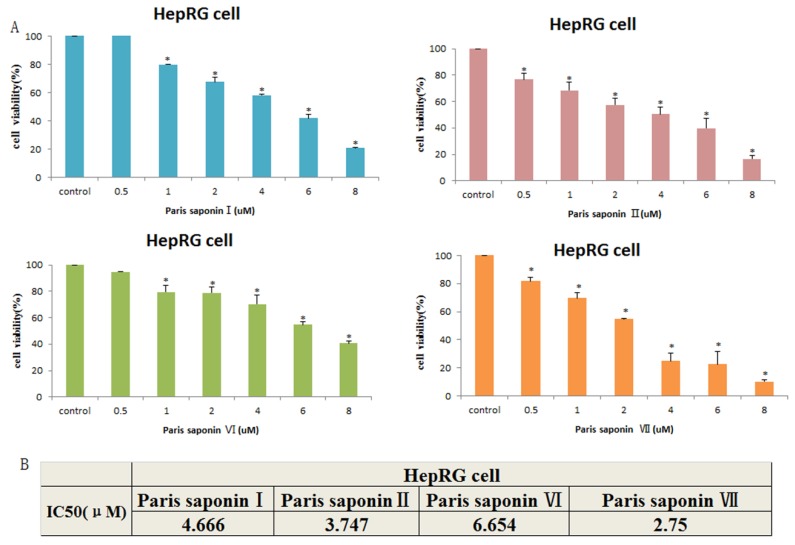
Cytotoxic effects of the four saponins on HepaRG cells. (**A**) Cell viability of HepaRG cells as measured with an MTT assay. Data are presented as mean ± SD. ** p* < 0.05, compared with the vehicle-treated group. (**B**) The cell viability data of HepaRG cells treated with four saponins for 24 h were analyzed with the SPSS17 software, and the IC_50_ values were obtained.

**Figure 4 cells-08-00690-f004:**
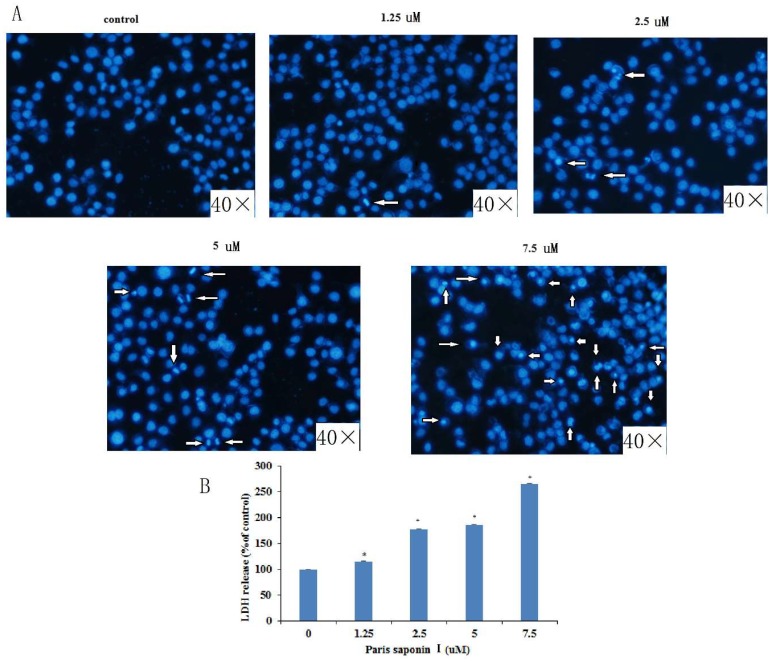
LDH release and DAPI staining. (**A**) Effect of *Rhizoma paridis* saponin (PSI) on nuclear morphology, as observed with an inverted Olympus IX71 fluorescence microscope after DAPI staining. (**B**) Effect of PSI on lactate dehydrogenase (LDH) leakage, as an index of the cytotoxicity of PSI.

**Figure 5 cells-08-00690-f005:**
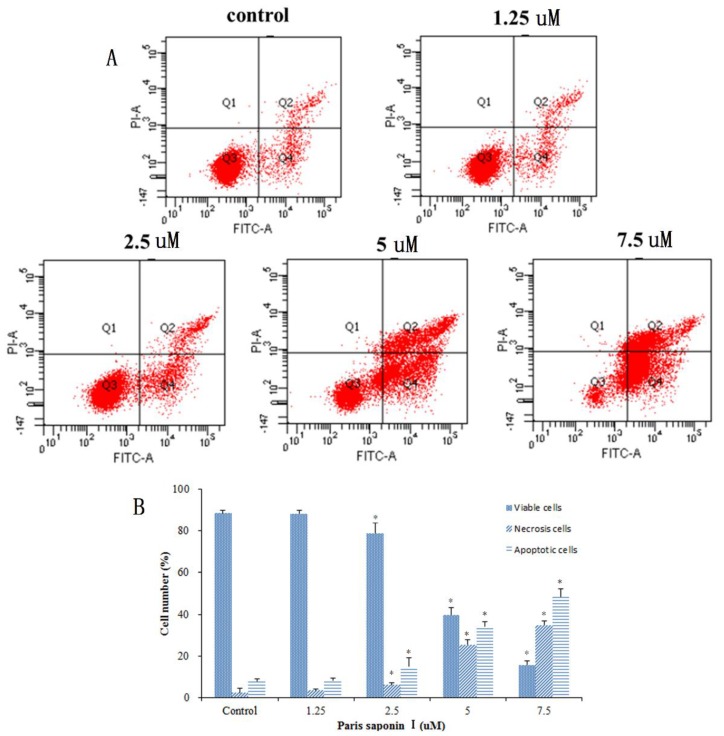
The effect of PSI on the apoptosis and necrosis of PSI. (**A**) Flow cytometric analysis of Annexin V and PI staining after treatment with PSI (0–7.5 uM) for 24 h. The lower right quadrant depicts early apoptotic cells; the upper right quadrant shows necrotic or late-apoptotic cells, while the Lower left quadrant shows viable cells. (**B**) The percentage of HepaRG cells in the survival, early apoptosis, late apoptosis, and necrosis categories. Data are displayed as mean ± SD. * *p* < 0.05, compared with the vehicle-treated group (0 μM).

**Figure 6 cells-08-00690-f006:**
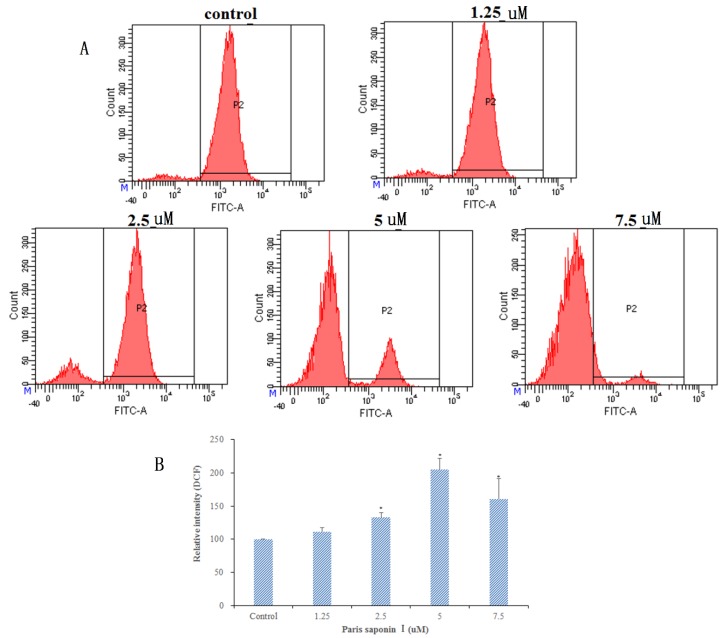
Effect of PSI on reactive oxygen species (ROS) generation in HepaRG cells after 24 h incubation. (**A**) Analysis of ROS generation using the oxidant sensitive fluorescent probe DCFH-DA. (**B**) The percentages of intracellular ROS after 24 h incubation with graded concentrations of PSI. Data are presented as mean ± SD. * *p* < 0.05, compared with the vehicle-treated group.

**Figure 7 cells-08-00690-f007:**
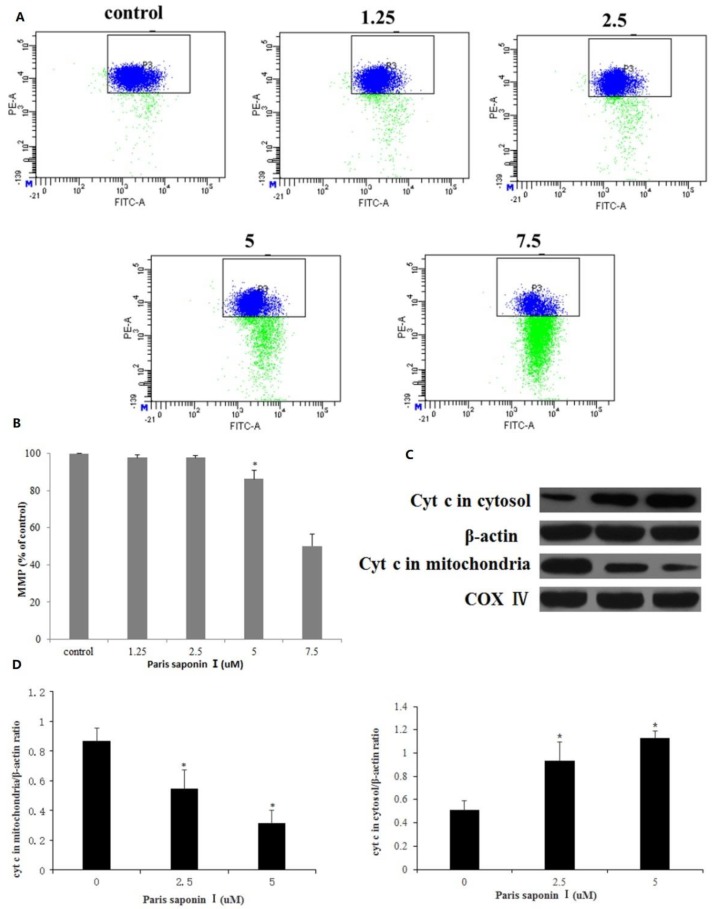
Effects of PSI on mitochondrial membrane potential (MMP) and the release of cytochrome c in HepaRG cells. (**A**) MMP was determined flow cytometrically. (**B**) The percentage of MMP in HepaRG cells. Data are presented as mean ± SD. * *p* < 0.05, compared with the control group. (**C**) The content of cytochrome c in mitochondria and cytoplasm, as determined using Western blot. Cytochrome c oxidase subunit 4 (COX IV) and β-Actin served as internal controls for the mitochondrial fraction and cytosolic fractions, respectively. (**D**) Optical density quantification results. Data are presented as mean ± SD. * *p* < 0.05 compared with the vehicle-treated group.

**Figure 8 cells-08-00690-f008:**
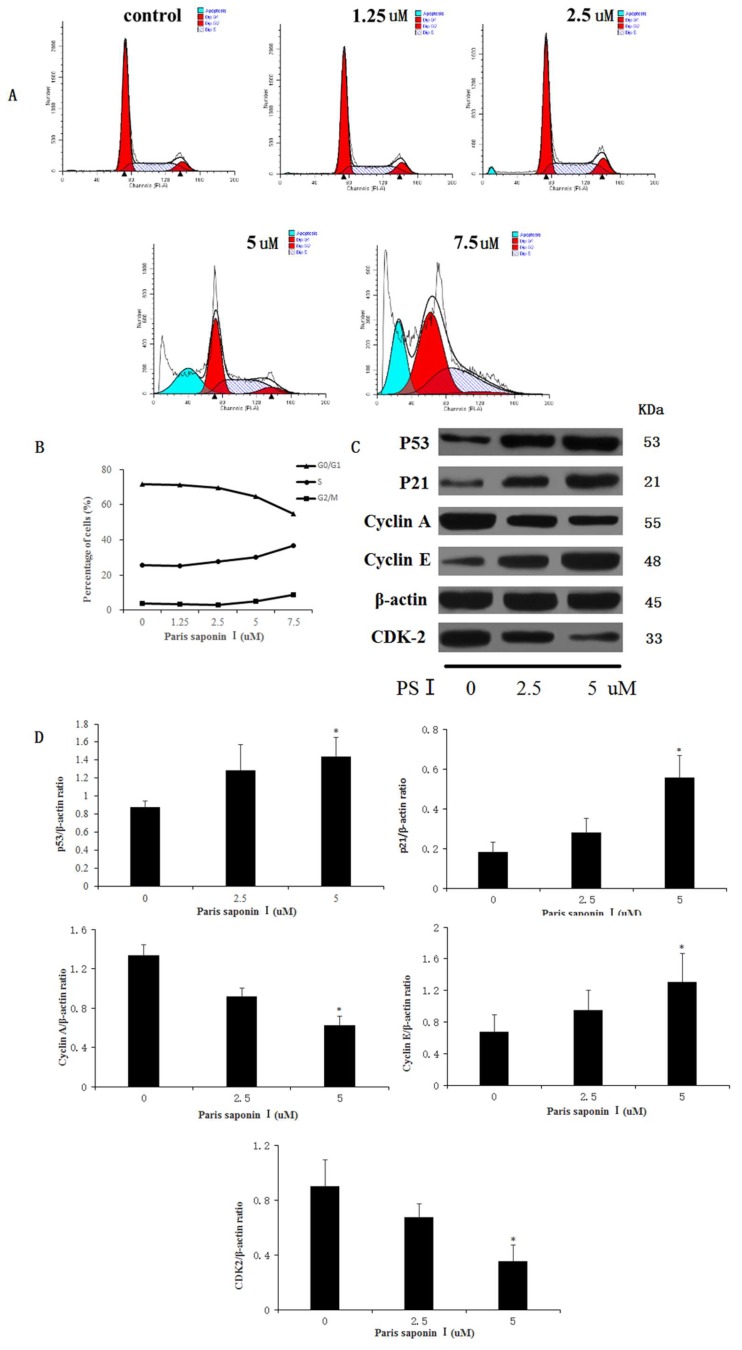
The effect of PSI on cell cycle. (**A**) HepaRG cell cycle distribution graphs for different groups, as measured with flow cytometry. (**B**) A broken line graph indicating the percentage of HepaRG cells in each phase of the cell cycle. (**C**) Protein expression levels of p53, p21, cyclinA, cyclin E, and CDK2, as determined using Western blotting. (**D**) Quantitative display of optical density values. Data are expressed as mean ± SD. * *p* < 0.05, compared with the vehicle-treated group.

**Figure 9 cells-08-00690-f009:**
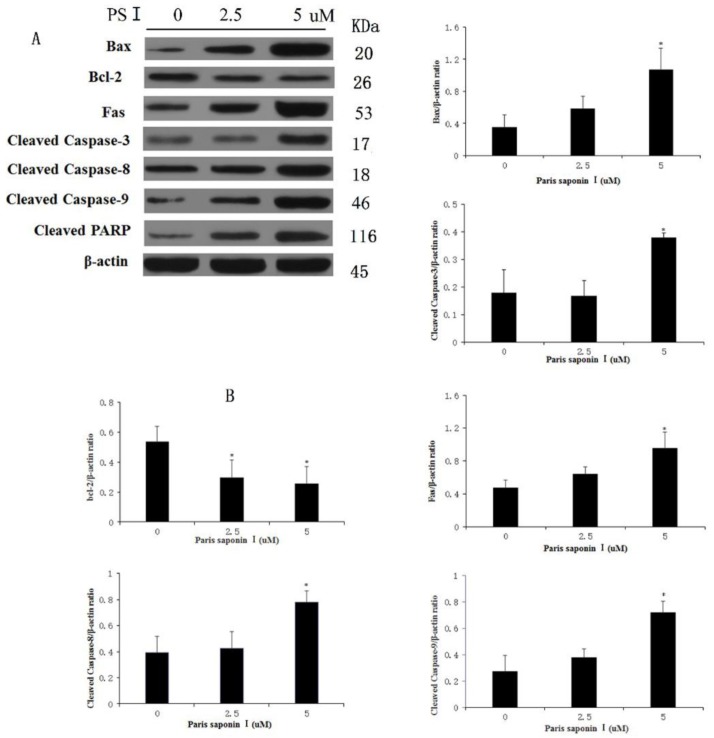
Effect of PSI on the expressions of apoptosis-related proteins in HepaRG cells. (**A**) Protein levels of Fas, Bax, Bcl-2, caspase-3, -8, -9 and PARP after treatment with PSI for 24 h, as determined with western blot analysis (**B**) Quantitative display of optical density values. Data are presented as mean ± SD of the three experiments. * *p* < 0.05, compared with the vehicle-treated group.

**Figure 10 cells-08-00690-f010:**
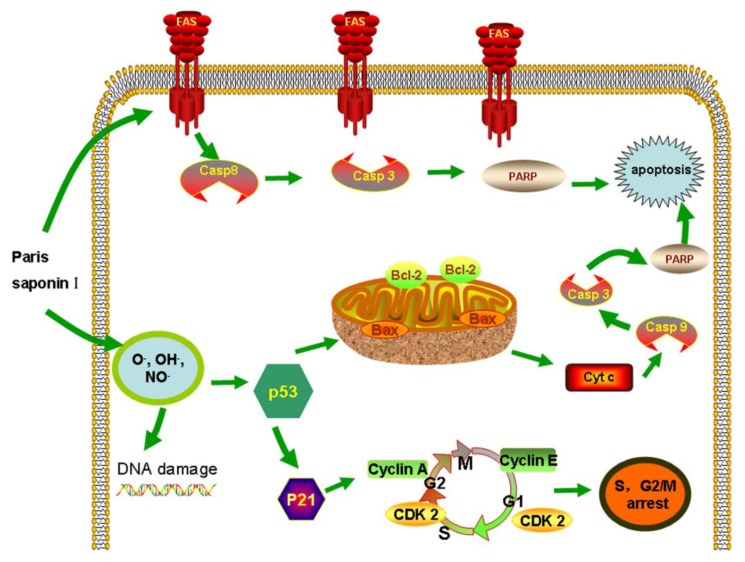
The extrinsic and intrinsic pathway of apoptosis induced by PSI.

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
