# Peer review of "Hepatocellular Toxicity of Paris Saponins I, II, VI and VII on Two Kinds of Hepatocytes-HL-7702 and HepaRG Cells, and the Underlying Mechanisms"

_cells, 2019, doi:10.3390/cells8070690_

Round 1
Reviewer 1 Report
-The authors have not provided information to justify the work done. What are the future applications of this knowledge? Authors should mention about the impact of this kind of studies in the field.
-I suggest to change the title in order to be more concise and specific.
-There are spelling and grammatical errors throughout the text (e.g., lines 27, 20, 223).
-I recommend to re-write the Introduction section because I found it some confuse. Please, authors should define the rationale of this work, and the aim is missing in introduction.
-I suggest to discuss their results accordingly. Only eight references (of 31) are used to discuss the results. Further studies should be included in the discussion section.
-References are not in the format required by the journal.
Author Response
In the discussion part, I added the significance of this study and its influence on future research.
I have changed the title of my article to make it more concise and specific.
I have used the "MedSciEdit" to re-edit and modify the whole text to make the language appropriate. And the manuscript ID is MSE06190375855.
I've rewritten the introduction to make it clearer and more logical. In addition, I have defined the rationale of this work and supplemented the purpose of the experiment.
The discussion part is supplemented and improved. In addition, references for discussion are added to make the basis of this study more sufficient.
I have modified the format of the references in the article to meet the requirements of the magazine.
Thank you very much for your efforts in this article.If you have any question about my article, please contact me at any time.
Reviewer 2 Report
Please address the following issues.
1) The manuscript requires some improvement in the quality of the written English. Please have it edited by a native English speaker, preferably a professional scientific proof-reader.
2) The structural drawings of paris saponins I, II, VI and VII should be corrected and improved. Please refer to PLOSONE|DOI:10.1371/journal.pone.0150595 March3,201.
3) The authors used commercially available paris saponins I, II, VI and VII throughout the experiments. However, their purities are not enough to evaluate biological activity. They should be purified by some chromatographic methods.
4) Paris saponins I, II, VI and VII are thought to exhibit cytotoxic activity against cultured tumor cell lines. The performed experiments have revealed the cytotoxic mechanisms of paris saponin I against HepaRG cells rather than paris saponins’ hepatotoxicity. The objective of the experiments should be reconsidered.
5) Line 117: [HepaRG] should be revised as [ HL-7702].
6) In Figure 10, the cascade that caspase 3 activates caspase 9 is not correct.
Author Response
1. I have used "MedSciEdit" to re-edit and modify the entire text to make the language appropriate. The original manuscript ID is MSE06190375855.
2.The structural drawings of paris saponins I, II, VI and VII have been revised according to the format of references.This figure is shown in fig1.
3.The standard products used in this study were all purchased from China's most authoritative standard product provider, China food and drug administration, and the content of standard products was over 97%, which could meet the requirements of the experiment.
4. Thank you for your Suggestions. According to your Suggestions, the experimental purpose of this paper has been revised to focus more on the hepatocellular toxicity of the four saponins rather than the hepatocellular toxicity of the four saponins, making this article more accurate.
5. The HepaRG have revised as HL-7702.
6.There are some inaccuracies in the previous pathway diagram, which have been modified, as shown in fig. 10.
Thank you very much for your effort on my anrticle.If you have any question, please contact me at any time.
Round 2
Reviewer 2 Report
The manuscript has been appropriately revised according to the suggestions and indications, and may be accepted for publication in Cells.